# The Moderating Effect of Personal Assets in the Relationships between Subjective Health, Housing Expense, and Life Satisfaction for Korean Middle and Old-Aged

**DOI:** 10.3390/healthcare11212866

**Published:** 2023-10-31

**Authors:** Kyung-A Sun, Joonho Moon

**Affiliations:** 1Department of Tourism Management, Gachon University, Seongnam 13120, Republic of Korea; kasun@gachon.ac.kr; 2Department of Tourism Administration, Kangwon National University, Chuncheon 24341, Republic of Korea

**Keywords:** Korean senior citizens, life satisfaction, subjective health, housing expense, personal assets, and moderating effects

## Abstract

The proportion of the elderly in Korea has increased. Given the circumstances, this research is intended to explore the characteristics of the elderly. The aim of this research is to explore the antecedents of life satisfaction in the Korean elderly using subjective health. Next, the goal of this research is to appraise the moderating effect of personal assets in the relationship between life satisfaction and housing expenses. The study data consist of 7199 observations from the 2018 and 2020 waves of the Korean Longitudinal Study of Aging (KLOSA). This research uses econometric analysis to test the research hypotheses, which include ordinary least squares, fixed-effects, and random-effects regression analyses. Following ordinary least squares, fixed-effects, and random-effects regression analyses, the results indicate a positive influence of subjective health on older Koreans’ life satisfaction. Additionally, personal assets positively moderate the association between housing expense and life satisfaction, the most valuable finding of the study. This research sheds light on the literature by revealing the moderating effect on the relationship between housing expense and life satisfaction. Moreover, the results could be used for better policy design with respect to the middle- and old-aged members Korean society.

## 1. Introduction

Korean society has been rapidly aging. According to Statistics Korea [1], Korean society’s current population older than age 65 is approximately 9.01 million, 17.5% of the total population. Statistics Korea [1] forecasted that the proportion of the older adults is to increase to 20.6% in 2025. Moreover, 43.3% of older adults had not adequately prepared for their financial futures, which has led to increasing poverty among the elderly in Korea [1]. This reality has serious economic implications for Korean society. In order to solve the problem, it is essential to understand the financial characteristics of older adults.

The main element of this research is life satisfaction, a common research focus for understanding individual behavior because it encompasses overall living conditions and well-being [2,3,4]. Moreover, researchers have also frequently selected life satisfaction as a dependent variable [3,5,6,7]. Based on numerous previous studies, it can be inferred that life satisfaction is worthy of investigation. Therefore, this research adopts life satisfaction as the main explanatory attribute.

Subjective health is the first element of life satisfaction that was investigated in this research. In old age, health is the most critical issue because poor health not only diminishes overall life quality but potentially hastens death [8,9,10]; that is, one’s health condition is likely to become the precondition for a better life. Thus, this research selects subjective health as the first element with which to account for life satisfaction. The second element of life satisfaction explored in this work is housing expense, which refers to the budget for administering housing. Statistics Korea [1] reported that Korea’s real estate sentiment index has fluctuated greatly in the last five years because of government policy. This instability has increased difficulties for Korea’s elderly because housing is usually their largest expense. Indeed, scholars have alleged that housing becomes more important for the elderly because they spend more time in their home compared to when they were young [11,12,13]. Such an argument inspired this research on the impact of housing expenses on life satisfaction in this population.

The last domain addressed in this work is personal assets. Under capitalism, personal assets confer a kind of social power [14,15]. Individuals with significant assets are likely to have better life conditions, including the elderly. Older adults have few opportunities to earn or improve their financial conditions, and senior citizens’ life quality likely varies depending on their personal assets. Given this condition, the final aim of this research is to inspect the moderating effect of personal assets between housing expense and life satisfaction and quality for the Korean elderly. This is because sufficient assets held by the elderly are likely to decrease the burden of their housing expenses.

All things considered, the main purpose of this research is to investigate the determinants of life satisfaction for older adults in Korea. As the determinant, this research first chooses subjective health. Moreover, another goal of this work is to examine the effect of housing cost and establish the moderating effect of personal assets on the relationship between housing cost and life satisfaction.

Even though housing costs are a critical issue in Korea, few researchers have explored their impact on the living conditions of the elderly. Additionally, there has been little examination in the extant literature of the moderating effects of personal assets on the relationship between housing expenses and life satisfaction among senior citizens. This research attempts to fill this gap in the research. Therefore, this research sheds light on the literature by disclosing the relationship between life satisfaction, subjective health, housing expense, and personal assets. An additional goal is to present policy implications based on the findings from this research. This information might become valuable as a reference for policy makers intending to allot government resources more efficiently.

## 2. Review of Literature and Hypotheses Development

### 2.1. Life Satisfaction

Life satisfaction refers to an individual’s contentment with their current life situation [2,3,4]. The extant literature has also demonstrated that life satisfaction is an indicator of overall individual living conditions based on personal evaluation [2,6,16]. According to St. John et al. [17], life satisfaction is likely to work as a predictor of mortality for older adults. Previous researchers have scrutinized the determinants of life satisfaction because it is a representative indicator of individual life quality [2,6,18]. To be specific, Omri et al. [19] explored the effect of energy consumption on individual life satisfaction. Vujić and Szabo [20] inspected the impact of smart phone use on life satisfaction. Moreover, Wang et al. [16] employed students and examined the determinants of life satisfaction. Chen et al. [2] also explored the characteristics of children’s life satisfaction. Furthermore, Numerous studies have investigated the life satisfaction of the senior citizens. For instance, Bidzan-Bluma et al. [5] studied older Polish and German adults and identified multiple influential variables for life satisfaction. Moon et al. [7] examined the determinants of life satisfaction by employing Korean middle- and old-aged citizens. Clair et al. [6] studied the impacts of social isolation on the life satisfaction of a population of Spanish elderly adults, and Matud et al. [21] studied the antecedents of life satisfaction by employing older adults. Moreover, Khodabakhsh [22] implemented a literature review to examine the characteristics of life satisfaction in the case of the elderly. The extensive literature on life satisfaction validates it as a worthy topic of study.

### 2.2. Subjective Health and Its Impact on Life Satisfaction

Subjective health is defined as an individual’s subjective assessment of their personal health conditions, including both mental and physical aspects [23,24,25]. Researchers have widely studied subjective health because the concept includes both psychological and physiological elements for the appraisal of individual health conditions [9,23,26]. Prior studies have, in fact, argued for good health as a precondition for better life and identified clear positive effects of subjective health on life satisfaction [7,21,27]. Wurm et al. [8] established that subjective health is positively associated with older adults’ life satisfaction, and Stephan et al. [10] identified a positive influence of subjective health on life satisfaction among the elderly. Additionally, Moon et al. [9] found a positive impact of subjective health on life satisfaction among older Korean adults. Plus, Mitra et al. [27] found that subjective health exerted a positive impact on the life satisfaction of employees across 53 countries. In a similar vein, Qazi et al. [28] inspected older women and the results indicated that subjective health is positively associated with life satisfaction. Furthermore, Stephan et al. [10] unveiled the positive effect of subjective health on life satisfaction by employing older adults. Plus, Fukahori et al. [29] revealed the positive impact of subjective health on the life satisfaction of middle-aged family members. Based on these findings of the previous literature, the following research hypothesis is proposed:

**Hypothesis** **1** **(H1).**
*An increase in subjective health results in increase in life satisfaction of the Korean elderly.*


### 2.3. Housing Expense and Its Impact on Life Satisfaction

Housing expense refers to any expenditures for housing: leases, taxes, maintenance, etc. [30,31]. Scholars also defined housing cost as the cost of maintaining the quality of housing for individual living [12,32,33]. Elderly people spend a great deal of time in their homes, so adequate housing is imperative for them [10,12,13,33]. In general, housing is a fixed cost, and higher housing costs reduce available funds for daily living expenses such as leisure, food, cultural and arts activities, and healthcare [32,34,35,36]. In fact, Vallerand et al. [37] documented that individuals with more financial distress due to a higher proportion of sunk cost in their living expenses showed a lower level of life satisfaction. Plus, Acolin and Reina [38] argued that high housing costs exert a negative effect on life satisfaction. Shim et al. [39] researched people with disability and the results demonstrated that a higher proportion of fixed cost negatively affected life satisfaction because individuals did not possess sufficient money for other areas to take care of their living. Based on the review of literature, the following research hypothesis is proposed:

**Hypothesis** **2** **(H2).**
*Increase in housing expense results in decrease in life satisfaction of the Korean eldelry.*


### 2.4. Personal Assets and Its Moderating Effect

Personal assets reflect a person’s personal wealth [40,41]. Previous works also defined personal assets as the amount personal resource with certain value [14,42,43]. Individuals possessing sufficient personal assets face few constraints on purchasing goods in their lives, including the payment of their housing expenses [41,43,44]. Moreover, scholars alluded that personal assets are a sort of power for living because economic goods and service could be attained by personal wealth [42,45]. In addition, extant literature documented that personal wealth is an indicator of individuals’ available resources which enables individual life to become better [14,15,45]. Thus, it can be inferred that having sufficient personal assets can offset the impact of housing expenditures on a person’s life satisfaction because financial power could minimize the damage caused by housing expenses. Based on these literature findings, the following research hypothesis is proposed:

**Hypothesis** **3** **(H3).**
*Personal assets positively moderate the relationship between housing expense and life satisfaction of the Korean elderly.*


## *3.* Method

### 3.1. Research Model and Data Collection

Figure 1 presents the research model with life satisfaction as the dependent variable. The independent variables are subjective health and housing expense, and the personal assets variable is the moderator between housing expense and life satisfaction. Subjective health exerts a positive effect on life satisfaction. However, housing expenses are negatively associated the life satisfaction. In addition, personal assets positively moderate the relationship between housing expense and life satisfaction.

The data came from the 2018 and 2020 waves of the Korean Longitudinal Study of Aging (KLOSA), which has been commonly employed to explore the behavioral characteristics of Korean senior citizens [46,47,48]. It popularity led this research to adopt KLOSA as the source of the data. KLOSA research panel data offer survey-based information on senior citizens across different periods [49]. The study period of this work is 2018–2020. The survey is performed every two years, and the data comprise the most updated version. The panel data were unbalanced and resulted in a total of 7199 observations. Unbalanced panel data mean that all participants did not take part in the survey during the study period [50].

### 3.2. Variable Description

Table 1 presents the set of variables and how they were measured: life satisfaction (LST), subjective health (SHE), housing expense (HOE), personal assets (PAS), gender (GEN), age (AGE), and COVID-19 (COV). LST ranges from 0 to 100, and the higher value stands for higher level of life satisfaction. Subjective health was measured on a five-point scale (1 = very poor; 5 = very good). The measurement of HOE was monthly housing expense over total monthly living expense. For PAS, it is measured by the amount of individual’s personal assets. GEN and AGE are the gender and physical age of survey participants, respectively. LST is the dependent variable; SHE and HOE are intendent variables; PAS is the moderator; GEN, AGE, and COV are control variables. This research used binary variables to measure COVID (corona virus diseases in 2019) because the infectious disease had a huge impact on individual life.

### 3.3. Data Analysis

This study implemented descriptive statistics analysis by computing mean, standard deviation (SD), minimum, and maximum. Next, correlation matrix analysis was implemented to examine the associations between variables. For hypothesis testing, this research executed econometric analysis for panel data: ordinary least squares (OLS), fixed effects (FE), and random effects (RE) [49,50]. Panel data analysis is critical for minimizing bias in estimation. OLS minimizes the errors in estimating coefficients [49,51]. FE analysis minimizes omitted variable bias by incorporating dummy variables to control time effects [50,51]. For FE analysis, this research incorporated the COV variable into the regression model to minimize the bias caused by the year effect. RE analysis entails entering unobserved effects into a model for estimating coefficients [49,50]. In this study, a moderating effect was tested by employing the interaction variable HOE × PAS. The following is the regression equation of this research:*LST_it_* = *β_0_* + *β_1_SHE_it_* + *β_2_HOE_it_* + *β_3_PAS_it_* + *β_4_HOE_it_* × *PAS_it_* + *β_5_GEN_it_* + *β_6_AGE_it_* + *ε_it_*,
where the following abbreviation apply: LST: life satisfaction; SHE: subjective health; HOE: housing expense; PAS: personal assets (Unit: 10,000 KRW); GEN: gender; AGE: age; i: ith participants; t: tth year; and *ε* is a residual.

Next, a median split was calculated to further test for any moderating effect of PAS. The median of HOE was 0.1, while the median of PAS was 20,000. Using the median value, this research produced four cases with the calculation of the mean values of LST. The graphical presentation enabled this research to scrutinize the moderating effect of personal assets.

## 4. Results

### 4.1. Descriptive Statistics and Correlation Matrix

Table 2 presents the descriptive statistics of the data. The number of observations is 7199. LST has a mean value 61.71, with 16.71 as the standard deviation. The mean value of SWE is 2.90, and its standard deviation is 0.85. For HOE, the mean value is 0.11, with 0.06 as standard deviation. Table 2 also describes the information of PAS (mean = 30,929.79; SD = 42,077.50; minimum = 0; maximum = 818,000), GEN (mean = 0.35; SD = 0.47), AGE (mean = 72.17; SD = 9.19; minimum = 57; maximum = 102), and COV (mean = 0.49; SD = 0.50).

Table 3 presents the correlation matrix. LST positively correlates with SHE (r = 0.245; *p* < 0.05), PAS (r = 0.221; *p* < 0.05), GEN (r = 0.046; *p* < 0.05), and COV (r = 0.030; *p* < 0.05), while LST negatively correlates with HOE (r = −0.145; *p* < 0.05) and AGE (r = −0.135; *p* < 0.05). SHE positively correlates with PAS (r = 0.105; *p* < 0.05) and GEN (r = 0.104; *p* < 0.05), whereas SHE negatively correlates with HOE (r = −0.109; *p* < 0.05) and AGE (r = −0.397; *p* < 0.05). HOE negatively correlates with PAS (r = −0.154; *p* < 0.05), GEN (r = −0.044; *p* < 0.05), and AGE (r = −0.085; *p* < 0.05). PAS positively and negatively correlates with GEN (r = 0.048; *p* < 0.05) and AGE (r = −0.085; *p* < 0.05), respectively. GEN negatively correlates with AGE (r = −0.030; *p* < 0.05), whereas GEN positively correlates with COV (r = 0.175; *p* < 0.05). Last, AGE positively correlates with COV (r = 0.105; *p* < 0.05).

### 4.2. Results of Hypotheses Testing

Table 4 is the results of regression analysis. All three models are statistically significant (*p* < 0.05). LST is the dependent variable. SHE is positively associated with LST (β = 4.17; *p* < 0.05). HOE negatively impacted on LST (β = −29.53; *p* < 0.05). PAS exerted a positive impact on LST (β = 5.80 × 10^−5^; *p* < 0.05). Next, LST is also positively affected by HOE × PAS (β = 1.90 × 10^−4^; *p* < 0.05). With regard to the results of regression analysis, all the proposed hypotheses are supported.

Table 5 presents the results of multiple linear regression analysis, including control variables. Model 4, Model 5, and Model 6 are OLS, RE, and FE, respectively. The intercept is significant (β = 53.64; *p* < 0.05). LST is the dependent variable. All three model are statistically significant in the 95% confidence interval (*p* < 0.05). SHE exerts positive impact on LST (β = 4.02; *p* < 0.05). HOE is negatively associated with LST (β = −28.78; *p* < 0.05). PAS exerts positive effect on LST (β = 5.77 × 10^−5^; *p* < 0.05). LST is also positively influenced by HOE × PAS (β = 1.91 × 10^−4^; *p* < 0.05). Given the results of regression analysis, all the proposed hypotheses are supported. The direction and significance appeared as consistent in all three models.

Table 6 and Figure 2 present the results of a moderating effect of personal assets. The elderly with low housing expenses and the rich showed the highest value in terms of life satisfaction. Moreover, Figure 2 presents that senior citizens with a high amount of housing expenditure and the poor reported the lowest value for life satisfaction. Table 6 shows the mean value of four groups (rich and low housing expense: 66.57; rich and high housing expense: 65.27; poor and low housing expense: 58.68; poor and high housing expense: 56.21).

## 5. Discussion

The purpose of this study was to explore the determinants of life satisfaction for the Korean elderly extrapolated from data from the 2018 and 2020 waves of the longitudinal KLOSA. The descriptive statistics showed that older Korean adults were spending an average of approximately 11% of their living expenses on housing. The results also unveiled that good subjective health exerted a positive impact on life satisfaction. To be specific, as one level of subjective health increased, the life satisfaction increased by approximately 4.17 (see Table 4). However, housing costs reduced elder adults’ life satisfaction, possibly because of their more limited resources in proportion to their housing expenditures. In addition, the results uncovered that senior citizens possessing more wealth showed the better life condition. The next aim of this work was to examine the moderating effect of personal assets on the relationship between housing expense and life satisfaction. The results implied that personal assets exerted a positive moderating effect on the association between housing expense and life satisfaction. However, the effect was not strong due to the unit of personal assets. In order to scrutinize the moderating effect of personal assets, this work implemented median split analysis using a personal assets median value of 20,000. Regarding the results, elderly Koreans with few personal assets reported a lower level of life satisfaction than did older adults with high personal assets. This suggests that housing expenses became a greater burden for older adults who possessed a relatively smaller amount of wealth.

## 6. Conclusions

### 6.1. Theoretical Implications

The results of this work shed light on the literature by disclosing the association between the housing expense and life satisfaction of the older adults. Because of the limited opportunity for earning in old age, this research presumed that a negative effect of housing expenses on life satisfaction of the elderly. Indeed, this study demonstrated a significant relationship between housing expenses and life satisfaction. This study also contributed to the literature by elucidating associations between life satisfaction, subjective health, housing expense, and personal assets, employing the Korean elderly as the research subject. In detail, this research demonstrated a positive and significant moderating effect of personal assets in the relationship between housing expense and life satisfaction among older Korean adults. Additionally, this research is theoretically worthy in that it reveals a significant link between housing expense and life satisfaction; the extant literature has shown sparse connections between these variables; thus, the outcome of this research partially bridges this research gap. In addition, this work ensured the relationship between subjective health and life satisfaction in the case of the Korean elderly. The results are aligned with the findings of the extant literature [8,9,21,26]; that is, this research externally validated the outcomes of prior studies.

### 6.2. Policy Implications

This study also has policy implications. First, policy makers might be able to consider devoting their resources to improving the health condition of senior citizen. This could be accomplished by investing more in the health insurance system and subsidizing medical expenses for senior citizens. Moreover, the subjective health condition might be able to be improved via cultural and outdoor activities because social interaction is likely to enhance mental health condition by minimizing isolation. Hence, policy makers might be able to direct their budget into the cultural, social, and recreational activities as well as healthcare system for senior citizens. Plus, policy makers might need to support the elderly by preparing housing-related policy, which could include subsidies, welfare policies, repairing housing facility, and offering cheap housing for the elderly. Moreover, policy makers could consider subsidizing housing expenses for elderly Koreans with few personal assets because of the significant negative impact of high housing expenses on their life satisfaction. Such a budget allocation would improve the life satisfaction of poorer elderly adults because it could reduce their financial burden with respect to housing. Such a policy might become the avenue by which to allot constrained resources more efficiently. However, the selection process needs to be implemented carefully to avoid inaccurate resource allocation. Next, the results from calculating the personal assets revealed that 200,000,000 KRW could become the median that divides low from high personal assets; policy makers could consider this a cutoff in determining criteria for building policy. Furthermore, the results showed that a higher level of personal assets is essential for a better life. Government policy might thus be able to focus more on how individuals prepare, financially, for their old age. This might be accomplished via the pension system. Moreover, policy makers might be able to dedicate the government resources to creating job positions for older adults because jobs play an imperative role in earning, which enhances the financial condition of the elderly. Furthermore, the results of this work could become a reference for countries with low fertility rates, which include Taiwan, Italy, Spain, and Singapore, because these countries have aj aging society [52].

### 6.3. Limitations and Suggestions for Future Works

This study does have limitations. First, the study was limited to examining only Korean senior citizens. In order to overcome such a limitation, future researchers might be able to examine other determinants of life satisfaction for the elderly using more international case. Moreover, life satisfaction was the only primary focus of this paper, although this research did present a couple of explanatory attributes. Therefore, the results of this research could be too limited to further our understanding of the behavioral characteristics of senior citizens. Thus, future researchers might be able to consider a broader set of variables by which to better understand the characteristics of the elderly. In addition, this research used archival data, and the measurement process was not delicate. Hence, future research might be able to explore and develop more diverse scales and demonstrate their effect. Such an effort might become an avenue by which to better understand the elderly. Furthermore, the life satisfaction of the elderly might be influenced by housing type and dependence. Future research might, therefore, consider such attributes to better understand the behavior of older adults.

## Figures and Tables

**Figure 1 healthcare-11-02866-f001:**
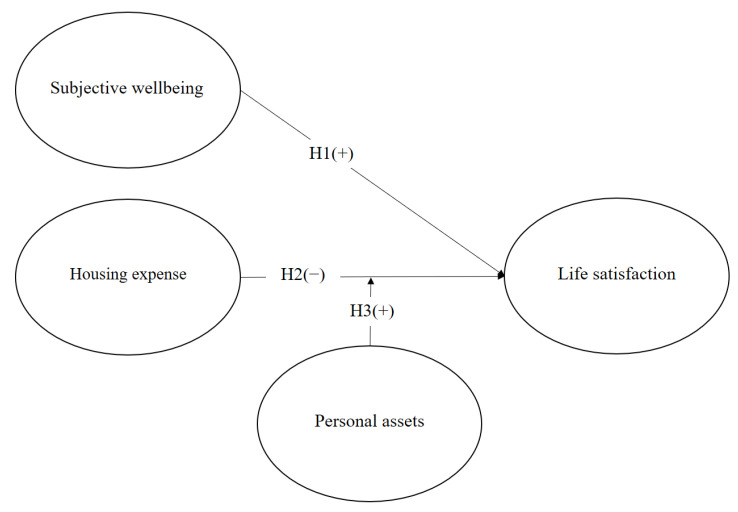
The research model.

**Figure 2 healthcare-11-02866-f002:**
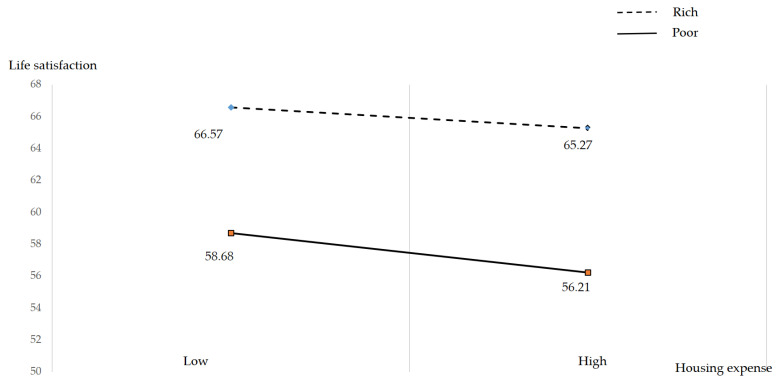
Moderating effect of personal assets.

**Table 1 healthcare-11-02866-t001:** Description of variables (N = 7199).

Variable	Code	Measurement
Life satisfaction	LST	Scale of 0 to 100 (0 = very unsatisfied, 100 = very satisfied)
Subjective health	SHE	1 = Very poor; 5 = Very good
Housing expense	HOE	Monthly housing expense/Total monthly living expense
Personal assets	PAS	Personal assets (10,000 KRW)
Gender	GEN	0 = Male; 1 = Female
Age	AGE	Physical age of survey participants
COVID-19	COV	0 = 2018; 1 = 2020

Note: KRW = Korean won.

**Table 2 healthcare-11-02866-t002:** Descriptive statistics (N = 7199).

Variable	Mean	SD	Minimum	Maximum
LST	61.71	16.71	0	100
SHE	2.90	0.85	1	5
HOE	0.11	0.06	0	0.625
PAS	30,929.79	42,077.50	0	818,000
GEN	0.35	0.47	0	1
AGE	72.17	9.19	57	102
COV	0.49	0.50	102	1

Note: LST: life satisfaction; SHE: subjective health; HOE: housing expense; PAS: personal assets (Unit 10,000 KRW); GEN: gender; AGE: age; SD: standard deviation.

**Table 3 healthcare-11-02866-t003:** Correlation matrix (N = 7199).

Variable	1	2	3	4	5	6	7
1. LST	1						
2. SHE	0.245 *	1					
3. HOE	−0.145 *	−0.109 *	1				
4. PAS	0.221 *	0.105 *	−0.154 *	1			
5. GEN	0.046 *	0.104 *	−0.044 *	0.048 *	1		
6. AGE	−0.135 *	−0.397 *	−0.085 *	−0.085 *	−0.030 *	1	
7. COV	0.030 *	−0.014	0.007	0.029 *	0.175 *	0.105 *	1

Note: * *p* < 0.05. LST: life satisfaction; SHE: subjective health; HOE: housing expense; PAS: personal assets (Unit 10,000 KRW); GEN: gender; AGE: age.

**Table 4 healthcare-11-02866-t004:** Results of hypotheses testing using main variables (N = 7199).

Variable	Model 1 OLS β (t-Stat)	Model 2 RE β (Wald)	Model 3 FE β (t-Stat)
Intercept	51.08 (58.73) *	51.08 (58.73) *	50.74 (56.44) *
SHE	4.17 (17.31) *	4.17 (17.31) *	4.19 (17.36) *
HOE	−29.53 (−8.63) *	−29.53 (−8.63) *	−29.44 (−8.60) *
PAS	5.80 × 10^−5^ (8.43) *	5.80 × 10^−5^ (8.43) *	5.78 × 10^−5^ (8.40) *
HOE × PAS	1.90 × 10^−4^ (2.55) *	1.90 × 10^−4^ (2.55) *	1.89 × 10^−4^ (2.53) *
COV	-	-	0.59 (1.49)
F-value	185.82 *	-	149.13 *
Wald χ^2^	-	743.29 *	-
R2	0.1082	0.1088	0.1084

Note: * *p* < 0.05. Dependent variable is LST; OLS: ordinary least square; RE: random effect, FE: fixed effect; LST: life satisfaction; SHE: subjective health; HOE: housing expense; PAS: personal assets (Unit 10,000 KRW).

**Table 5 healthcare-11-02866-t005:** Results of hypotheses testing with control variables (N = 7199).

Variable	Model 4 OLS β (t-Stat)	Model 5 RE β (Wald)	Model 6 FE β (t-Stat)
Intercept	53.64 (24.88) *	53.64 (24.88) *	53.68 (24.90) *
SHE	4.02 (15.36) *	4.02 (15.36) *	4.02 (15.37) *
HOE	−28.78 (−8.32) *	−28.78 (−8.32) *	−28.63 (−8.28) *
PAS	5.77 × 10^−5^ (8.37) *	5.77 × 10^−5^ (8.37) *	5.74 × 10^−5^ (8.34) *
HOE × PAS	1.91 × 10^−4^ (2.55) *	1.91 × 10^−4^ (2.55) *	1.91 × 10^−4^ (2.55) *
GEN	0.34 (0.82)	0.34 (0.82)	0.25 (0.60)
AGE	−0.03 (−1.34)	−0.03 (−1.34)	−0.03 (−1.52)
COV	-	-	0.63 (1.57)
F-value	124.29 *	-	106.91 *
Wald χ^2^	-	745.75 *	-
R2	0.1091	0.1091	0.1085

Note: * *p* < 0.05. Dependent variable is LST; OLS: ordinary least square; RE: random effect; FE is fixed effect; LST: life satisfaction; SHE: subjective health; HOE: housing expense; PAS: personal assets (Unit 10,000 KRW); GEN: gender; AGE: age.

**Table 6 healthcare-11-02866-t006:** Results of moderating effect by median split.

	Low Housing Expense	High Housing Expense
Rich	66.57	65.27
Poor	58.68	56.21

Note: Dependent variable is LST (life satisfaction). PAS_Median_ = 20,000.

## Data Availability

Not applicable.

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
