# Peer review of "The Moderating Effect of Personal Assets in the Relationships between Subjective Health, Housing Expense, and Life Satisfaction for Korean Middle and Old-Aged"

_healthcare, 2023, doi:10.3390/healthcare11212866_

Round 1

Reviewer 1 Report

Comments and Suggestions for Authors

The article addresses an important problem, but in my opinion the paper requires significant changes and additions:

1. The abstract should be supplemented with research conclusions.

2. In my opinion, one main goal of the work and specific goals should be indicated. It is worth presenting them in a separate paragraph of the work.

3. In my opinion, the research hypotheses formulated in the paper are unverifiable. No directions of relationship between variables (changes in both variables, e.g. an increase in one variable results in a decrease in the other variable).

4. Lack of definition of the concepts discussed in the paper, e.g. subjective health, housing expense, life satisfaction.

5. There is no discussion of the results. No attempt was made to explain the results.

6. It is worth discussing the obtained models and their parameters in greater detail. Are they logical? Do values of determination coefficients slightly above 10% indicate limited quality of the models? Similarly, the correlation coefficients are statistically significant but close to 0.

7. Results, discussion, and conclusions are limited. It is worth expanding these parts of the paper, especially discussion and conclusions.

In my opinion, the paper requires significant corrections and additions. After the corrections, it should be re-evaluated, in my opinion.

Reviewer 2 Report

Comments and Suggestions for Authors

line 9: ..elderly in Korea has kept grown increased. Given the condition current situation, this research...

lines 75-76: A number of prior researchers have Previous research in this area has...  

1) The study focuses on the relationship among health, income, and affordable housing for "elderly" Koreans. How do middle-aged Koreans factor into the study?  

2) Is there a distinction among different types of housing needs relative to health and age where transitions from independent to dependent living may be required?

3) Is intergenerational care given by family a factor in Korean culture?  If so, what imapact does it have on housing affordablity, health and quality of life?

4) Even though the study focuses on Korea, does it have implications for other countries facing a similar increase in life-expectancy relative to declining birth rates?

Comments on the Quality of English Language

English language is okay overall.  See Comments for authors for specific comments.  Editing of english is recommended. 

Round 2

Reviewer 1 Report

Comments and Suggestions for Authors

The article improved by the authors has gained value, in my opinion. The authors responded to my previous comments. They made satisfactory corrections to the paper.

The paper still requires minor corrections, mainly of a technical nature.

I am not an English language expert, but in my opinion the article requires language correction. Please consider this.

After corrections, the paper can be published.

Author Response

Given the comments of reviewer, we did check the English one more time.